# Inhibition of Cell Apoptosis by Apicomplexan Protozoa–Host Interaction in the Early Stage of Infection

**DOI:** 10.3390/ani13243817

**Published:** 2023-12-11

**Authors:** Liyin Lian, Qian Sun, Xinyi Huang, Wanjing Li, Yanjun Cui, Yuebo Pan, Xianyu Yang, Pu Wang

**Affiliations:** 1Key Laboratory of Applied Technology on Green-Eco-Healthy Animal Husbandry of Zhejiang Province, Provincial Engineering Research Center for Animal Health Diagnostics & Advanced Technology, Zhejiang International Science and Technology Cooperation Base for Veterinary Medicine and Health Management, China Australia Joint Laboratory for Animal Health Big Data Analytics, College of Animal Science and Technology & College of Veterinary Medicine, Zhejiang A & F University, Hangzhou 311300, China; 202127020321@stu.zafu.edu.cn (L.L.); sq520112233@163.com (Q.S.); xinyihuang001@163.com (X.H.); lwj13627727828@163.com (W.L.); cuiyanjun@zafu.edu.cn (Y.C.); xianyu_yang@hotmail.com (X.Y.); 2Gansu Polytechnic College of Animal Husbandry and Engineering, Wuwei 733006, China

**Keywords:** apicomplexan protozoa, apoptosis, interaction

## Abstract

**Simple Summary:**

Protozoa that parasitize within host cells can avoid direct damage from the host immune system. In this review, we have revealed some key factors and signaling pathways that inhibit host cell apoptosis, regulate cell apoptosis signal transduction, and interfere with the molecular mechanisms of cell apoptosis. These findings provide important clues to further understand the infection mechanism and pathogenesis of Apicomplexan protozoa and a theoretical basis for the development of new therapeutic strategies and preventive measures.

**Abstract:**

Apicomplexan protozoa, which are a group of specialized intracellular parasitic protozoa, infect humans and other animals and cause a variety of diseases. The lack of research on the interaction mechanism between Apicomplexan protozoa and their hosts is a key factor restricting the development of new drugs and vaccines. In the early stages of infection, cell apoptosis is inhibited by Apicomplexan protozoa through their interaction with the host cells; thereby, the survival and reproduction of Apicomplexan protozoa in host cells is promoted. In this review, the key virulence proteins and pathways are introduced regarding the inhibition of cell apoptosis by the interaction between the protozoa and their host during the early stage of Apicomplexan protozoa infection. It provides a theoretical basis for the development of drugs or vaccines for protozoal diseases.

## 1. Introduction

Apicomplexa protozoa, including *Toxoplasma gondii* (*T. gondii*), *Plasmodium*, *Cryptosporidium*, *Neospora caninum* (*N. caninum*), *Eimeria*, etc., are obligatory intracellular parasites, often zoonotic pathogens, posing a considerable threat to economic development and public health [1,2]. Although apicomplexan parasites can evade direct attacks by the host immune system, infected cells still have the ability to restrain their proliferation and development through apoptosis [3]. The inhibition of apoptosis is, therefore, one of the main mechanisms by which the parasites ensure their growth, development, and reproduction [4]. It has been reported that *T. gondii* suppresses the apoptosis of T cells and macrophages during infection and, thereby, achieves persistent infection [5]; the *Theileria* and *Cryptosporidium* species can also inhibit host cell apoptosis [5]. Similarly, the infection of *Eimeria acervulina* (*E. acervulina*) merozoites inhibits the apoptosis of bovine kidney cells [6], and second-generation merozoites of both *Eimeria necatrix* (*E. necatrix*) and *Eimeria tenella* (*E. tenella*) can inhibit host cell apoptosis by activating the NF-κB signaling pathway [7]. Due to the difference of these parasites in their mechanism of host cell entry and final intracellular localization, the specific mechanism of Apicomplexan parasites’ inhibition of host cell apoptosis remains to be further explored.

## 2. Apoptosis

Apoptosis is an important host self-defense mechanism that plays key regulatory roles in maintaining normal tissue development, removing damaged and aged cells, and maintaining the functions of the immune system [8]. Apoptosis also plays an important role in the infectious process of pathogens. As a host defense mechanism, it is of great significance, especially in resisting infections by protozoa.

The regulation of cell apoptosis involves multiple signal transduction pathways, mainly the endogenous pathway (mitochondria-mediated pathway) and the extrinsic pathway (death receptor-mediated pathway) [9]. The former is mainly affected by intracellular damage signals and stress factors, while the latter is activated through the binding of extracellular death receptor ligands. The endogenous pathway is the most common and involves the mitochondria [7]. It mainly includes the following steps: firstly, external or damage stimuli induce changes in the expression level of Bcl-2 family proteins inside the mitochondria, leading to the increased permeability of the mitochondrial outer membrane [7] and, thus, to the release of mitochondrial proteins such as cytochrome C into the cytoplasm; secondly, cytochrome C in the cytoplasm binds to an apoptotic protease-activating factor (Apaf-1) to form a complex called the apoptosome, which activates procaspase-9 to be caspase-9 [10,11]; thirdly, the activated caspase-9 further activates other caspases family members (caspase-7, caspase-6, and caspase-3), initiating a cascade in the cell which cleaves specific intracellular proteins (PARP (PolyADP-Ribose Polymerase), GDID4, DFF-45 (DNA fragmentation factor-45), lamin A, keratin 18), ultimately leading to the death of the cell [11]; finally, the condensation of chromatin and DNA fragmentation occur in the nucleus [12]. The exogenous pathway is a process of cell apoptosis mediated by transmembrane receptors such as tumor necrosis factor-α (TNF-α) and the Fas ligand [13]. When there are corresponding ligand molecules outside the cell, the receptor proteins on the cell membrane are recruited together and bind to the ligand, forming a death-inducing signal complex, catalyzing and activating procaspase-8, and, finally, leading to the executive phase when cell apoptosis is activated [14,15]. Caspase activation triggers a series of apoptotic events, including cytoskeleton reorganization, cell membrane changes, enzymatic hydrolysis, and DNA degradation. Eventually, the cells fragment into apoptotic bodies that are taken up and phagocytosed by surrounding macrophages to complete the process of cell clearance [16].

## 3. Apicomplexan Protozoa Inhibit Cell Apoptosis in the Early Stage of Infection

*Plasmodium* inhibits host cell apoptosis by Caspase activity, the PI3K signaling pathway, and the regulation of apoptosis-related proteins’ expression [17]. *Plasmodium* sporozoites pass through and damage a number of hepatocytes before they enter into the last one. This damage leads to the release of HGF, which is the initial reaction protecting the final host cell from spontaneous apoptosis. Since HGF/MET signaling is capable of protecting cells from apoptosis using both the PI3-kinase/Akt and MAPK pathways, the inhibition of HGF/MET signaling induces a specific increase in the apoptosis of infected cells and leads to a great reduction in infection; to counteract this, the infected hepatocytes express the HGF receptor c-met, which ensures the initial survival of the cells [18,19]. Furthermore, apoptosis is induced in mammalian cells in vitro using peroxide treatment or serum, but *Plasmodium* can inhibit cell apoptosis in vitro with peroxide treatment or serum deprivation and in vivo using TNF-α. This parasite-dependent inhibition of apoptosis is clearly detectable in vitro 24 h p.i. but is more pronounced 48 h p.i. At the latter time point, infected cells have been found to resist prolonged peroxide exposure [20].

This study suggests that *C. parvum* inhibits apoptosis by activating NF-κB signaling pathway during infection [21]. Interestingly, *C. parvum*-infected cells exhibit some signs of apoptosis after invasion during the early stages of infection [22]. Together, these observations suggest that *C. parvum* infection may interfere with the temporal progression of cell apoptosis and delay it sufficiently to support parasite replication.

Several strategies are also adopted to inhibit cell apoptosis for *T. gondii*, such as the regulation of the NF-κB signaling pathway and mitochondria-mediated apoptotic pathway. As an important anti-apoptotic factor, NF-κB is involved in the regulation of cell survival. *T. gondii* can inhibit cell apoptosis by activating NF-κB signaling pathway [11,23]. During the early infection of *T. gondii*, the inhibition of cell apoptosis also results from decreasing the expression of pro-apoptotic proteins and increasing the expression of anti-apoptotic proteins, thereby preventing mitochondria-mediated cell apoptosis [7].

In the early stage of infection of *Eimeria* species, the opening of mitochondrial permeability transition pore (MPTP) and the decrease in transmembrane potential were activated [24]; the Akt and ERK proteins, key factors of the epidermal growth factor receptor (EGFR) signaling pathway, were upregulated [25]. All these events synergistically contribute to the occurrence of anti-apoptosis.

Apicomplexan protozoa employ different strategies to inhibit apoptosis in order to protect themselves from host immune attacks and achieve survival in host cells. Therefore, these inhibitory mechanisms are important for further understanding the underlying infection process of protozoa.

## 4. Virulence Factors Associated with Apicomplexan Protozoa Inhibit Host Cell Apoptosis

### 4.1. Microneme Proteins

Micronemes are the special organelle located at the anterior end of apicomplexan. It secretes a variety of microneme proteins (MICs) and plays a key role in the life cycle of parasites, including invasion, immune evasion, and localization in host cells [26,27]. *E. tenella* MIC4 protein contains an EGF motif, which can interact with EGFR and activate the EGFR-Akt signaling pathway, finally inhibiting host cell apoptosis [28]. Similarly, *E. acervulina* MIC 3 protein can inhibit apoptosis of the chicken duodenal epithelial cell by interacting with the casitas B-lineage lymphoma (CBL) protein. The over-expression of *E. acervulina* MIC 3 protein increased the level of CBL expression, which mediated the ubiquitination of caspase 8 and promoted its degradation. The decrease in caspase 8 expression level finally led to the inhibition of cell apoptosis [6]. It has been shown that *T. gondii* can activate the EGFR-Akt signaling pathway through the EGF domain-containing MIC3, MIC6, and MIC8, thereby inhibiting host cell apoptosis [29,30,31].

### 4.2. Rhoptry Proteins

Rhoptry is a unique secretory organelle in protozoa. It has a rod-like outer shape and secretes proteins when the apicomplexan protozoa, such as *T. gondii*, *Eimera*, and *Cryptosporidium* species, invade host cells [1]. According to the localization, the secreted proteins are divided into rhoptry neck proteins (RONs) and rhoptry bulb proteins (ROPs), which are closely related to the invasion and inhibition of cell apoptosis, formation of the parasitophorous vacuole (PV), and formation of the moving junction (MJ) [32].

Rhoptry proteins inhibit apoptosis by affecting the Janus Kinase-signal transducer and activator of transcription (JKT-STAT), NF-κB, and MAPK signaling pathways in host cells [33]. For instance, ROP16 is a soluble rhoptry protein secreted by *T. gondii*. It was released in the host cell cytoplasm and then translocated into the cell nucleus through its nuclear localization structure (NLS), thereby affecting the expression of host genes [34]. ROP16 has tyrosine kinase activity and can activate the STAT3 and STAT6 signaling pathways by phosphorylating both Tyr705 and Tyr641 of the STAT3 and STAT6 complexes, thereby inhibiting cell apoptosis. In agreement with this idea, ROP16 promotes the synthesis of arginase 1, a major downstream substrate of the STAT6 signaling pathway in host cells [35]. In addition, another rhoptry protein ROP18 is shown to phosphorylate P65 and promote the degradation of p65 by the ubiquitin-dependent degradation pathway, suppress the NF-κB activation, and finally inhibit the production of proinflammatory cytokines and the apoptosis of host cells [36].

### 4.3. Dense Granule Proteins

Dense granule proteins (GRAs) have a diameter of about 200 nm and are distributed throughout the entire parasite. Approximately 70 GRAs have been reported so far, mainly including the typical GRAs, the inhibitors of STAT transcription (IST), mitochondrial association factors (MAF), and c-Myc regulation (MYR). GRAs related to apoptosis were observed in *Toxoplasma*-infected cells, such as GRA24 [37,38]. *T. gondii* GRA24 can induce higher activation of the subunit p38-alpha MAPK.

GRAs can inhibit apoptosis to achieve immune evasion and long-term parasitism [39]. It was shown that IST (a dense granule protein) secreted by *T. gondii* crossed the parasitophorous vacuole membrane (PVM) and accumulated in the nucleus to block the type I interferon response and promote the infection [40]. GRA5 can inhibit apoptosis of the infected cells to protect the parasites [41]. *T. gondii* GRA5 deletion mutant protects hosts against *T. gondii* infection, but the mechanism is unknown. Similarly, GRA24 is secreted by *T. gondii* into host cell cytoplasm and binds to p38 MAPK, activating GPCR/ PI3K/AKT signaling pathways, which induces p38 phosphorylation. The inhibitory effect of GRA24 on cell apoptosis needs to be further researched [42,43]. GRA15 interacts with TNF receptor-associated factors (TRAFs), which are adaptor proteins functioning upstream of the NF-Κb transcription factor. The deletion of TRAF-binding sites in GRA15 greatly reduces its ability to activate the NF-κB pathway, and TRAF2 knockout cells have impaired GRA15-mediated NF-κB activation [44]. Additionally, previous studies have reported that *T. gondii* also exploits heterotrimeric G protein-coupled receptor (GPCR)-mediated signaling to activate phosphoinositide 3-kinases (PI3Ks), leading to the phosphorylation of PI3K and MAPK pathways and inhibition of apoptosis [45].

### 4.4. Heat Shock Proteins

This is an evolutionarily conserved family of heat shock proteins. Based on molecular weight and sequence homology, heat shock proteins (HSPs) are classified into different families, including small HSPs, HSP40, HSP60, HSP70, HSP90, and large HSPs, and each family has a different intracellular distribution and functions [46]. HSPs have a variety of functions in organisms, especially in responding to external stresses, such as high temperature, oxidative stress, radiation, and infection [47]. HSPs also play a crucial role in preventing protein aggregation and promoting proteolysis by catalyzing the refolded damaged or denatured proteins [47]. In addition, they participate in the process of inhibiting cell apoptosis [48]. HSP70 is the main anti-apoptotic protein in *T. gondii*, which inhibits cell apoptosis via cytochrome c release, anti-apoptotic protein expression, and caspase inhibition [49]. The study demonstrated that the exertion of HSP70 anti-apoptotic effects is directly associated with the pro-apoptotic factors, APAF-1 and Apaf-1. HSP70 can bind to AIF (apoptosis-inducing factor) and apoptosis protease activate factor 1 (APAF-1), overexpress anti-apoptotic Bcl-2, inhibit AIF translocation and cytochrome C redistribution, and then inhibit cell mitochondrial apoptosis [50].

The above research has tried to describe the pathways that are controlled by the parasite to permit its survival inside the host. The main factors participating in the inhibition of cell apoptosis induced by protozoa are members of the MIC, ROP, HSP, and GRA family (Figure 1).

## 5. Apicomplexan Protozoa Infection Effects on Key Apoptotic Proteins and Pathways in the Host

### 5.1. Cysteine Protease Family

As a key component of the apoptosis signaling pathway, caspases are one class of cysteine proteases that hydrolyze the ASP-X peptide bond of substrate proteins [8]. They are divided into two groups: apoptotic initiating enzymes, such as Caspase-8 and Caspase-10, and effector enzymes, such as Caspase-3 and Caspase-7. The former initiates a cascade of reactions by activating effector enzymes, which eventually lead to changes in the morphological and biochemical characteristics of cells. However, the latter directly acts on a variety of intracellular substrates, such as nucleases and structural proteins, destroying the integrity and function of cells. Even if the apoptotic pathways are different, the Caspase cascade is a common event that is located downstream of the apoptotic signal transduction [51].

### 5.2. Bcl-2 Protein Family

Bcl-2 family is a group of proteins that regulate apoptosis and include members that inhibit apoptosis, such as Bcl-2 and Bcl-xL, and members that promote apoptosis, such as Bax and Bak [52]. These proteins are involved in apoptosis by regulating mitochondrial membrane permeability. Therefore, the inhibitors of Bax and Bak can prevent the release of cell death signaling molecules from mitochondria, thereby suppressing apoptosis [52].

Bcl-2 and Bax genes are the upstream factors of the apoptosis pathway. Under normal conditions, both form heterodimers to keep cells in a steady state. When they are stimulated by apoptotic signals, Bax is dissociated from heterodimers and forms the homodimer to induce cell apoptosis. Therefore, the ratio of Bcl-2/Bax in cells is an important factor in determining whether the cells will execute apoptosis after a certain stimulus [50]. The study demonstrated that the extract of Ashwagandha showed that the increase in the Bcl-2/Bax ratio could inhibit the cell apoptosis of splenic lymphocytes of treated mice. Similarly, *Eimeria* merozoite infection inhibited the host cell apoptosis by regulating the expression of Bcl-2 and Bax [53]. Compared with Bcl-2, Bcl-xl exhibits a stronger anti-apoptotic effect by preventing the release of cytochrome C and other apoptosis-inducing factors from mitochondria. On the contrary, Bid promotes apoptosis by increasing the release of those mitochondrion-derived apoptosis-inducing factors [45]. During *Eimeria* infection, the inhibition of host cell apoptosis was positively correlated with the ratio of Bcl-xl/Bid [53]. These results suggest that the *Eimeria* species inhibit apoptosis by preventing the dissociation of both Bcl-xl/Bid and Bcl-2/Bax heterodimers in host cells during infection.

The Bcl-2 gene family, Survivin genes, and microRNAs are involved in regulating host epithelial cell apoptosis induced by *C. parvum* infection. *C. parvum* upregulates the expression of anti-apoptotic genes and downregulates the expression of pro-apoptotic genes in the early stage (6–12 h) of infection with HCT-8 cells, while it upregulates the expression of pro-apoptotic genes and downregulates the expression of anti-apoptotic genes in the later stage (24–72 h) of infection. After silencing the Bcl-2 gene, cell apoptosis increases, indicating that the Bcl-2 gene family regulates host epithelial cell apoptosis induced by *C. parvum* infection [22]. A recent study suggests that in vitro infection of HCT-8 cells with *C. parvum* can induce the upregulation of the expression of the anti-apoptotic protein Survivin, thereby inhibiting the activities of caspase-3 and caspase-7, and finally weaken cell apoptosis. However, after silencing Survivin expression, cell apoptosis significantly increases and the number of parasites decreases; this indicates that the expression of Survivin can be regulated to inhibit cell apoptosis and promote the development of the parasite itself during *C. parvum* infection [23].

### 5.3. Cytochrome-C

Cytochrome-C is an essential component of the cellular respiratory chain and it participates in electron transfer [54]. Cytochrome-C causes cell apoptosis mainly in a Caspases-dependent manner. When cells receive apoptotic signals that cause mitochondrial dysfunction, Cytochrome-C is released from the mitochondrial membrane into the cytoplasm. Subsequently, Cytochrome-C first binds to Apaf-1 and causes conformational changes in Apaf-1, and then seven Apaf-1 molecules aggregate to form an apoptotic complex with the participation of ATP or dATP. This complex activates downstream apoptotic pathways, ultimately leading to cell death [55,56]. It has been found that the level of Cytochrome-C after *T. gondii* infection is significantly reduced, indicating that *T. gondii* infection inhibits cell apoptosis via the regulation of Cytochrome-C release (Table 1) [57].

Previous studies have shown that *T. gondii* RH strain tachyzoites in MEFs (Mouse Embryonic Fibroblasts) and *T. gondii* NTE strain tachyzoites in human promyelocytic leukemia cells (HL-60) can inhibit the release of mitochondrial cytochrome C by caspases 3 and 9. Further, after *T. gondii* infects human monocyte-infected cells (TH1), it can upregulate the anti-apoptotic protein Bcl-2 expression and downregulate the expression of pro-apoptotic protein Bax, thereby inhibiting mitochondrial permeability transition pore and blocking the release of cytochrome c from mitochondria.

Cryptosporidium can promote host cell apoptosis by releasing cytochrome C in the later stage of infection. Studies have evaluated the apoptotic changes in Cryptosporidium-infected cells in immunoactive (IC) and immunosuppressive (IS) mice either alone or loaded with silver nanoparticles (AgNPs) and NTZ. The treatment of NTZ loaded with AgNP at different doses showed the highest reduction in egg sac shedding and significant improvement in histopathological changes. Moreover, the levels of cytochrome C and caspase-3 were significantly reduced in IC and IS mice treated with NTZ loaded with AgNP, indicating that NTZ loaded with AgNP can inhibit cell apoptosis infected with Cryptosporidium by blocking the release of cytochrome C, which is a potential target for the treatment of neosporidiosis.

### 5.4. NF-κB Signaling Pathway

There are five members of the NF-κB family: P50 (NF-κB1), P52 (NF-κB2), P65 (RelA), RelB, and C-Rel [58]. NF-κB is an important transcription factor involved in the regulation of cell survival and apoptotic processes. NF-κB is a dimer of two subunits p50 and p65 [58] and, when not activated, it is bound to IκB in the cytoplasm. In response to certain stimuli, IκB is phosphorylated and degraded, leading to NF-κB transportation into the nucleus and thus the regulation of gene transcription [59].

It has been shown that apicomplexan parasites can inhibit host cell apoptosis by activating the NF-κB signaling pathway [60]. The activation of NF-κB can promote the expression of apoptosis-inhibiting factors and inhibit the production of apoptosis-related factors, thereby inhibiting cell apoptosis. Recently, all the NF-κB, phosphorylated NF-κB (p-IκBα), and Bcl-xL were demonstrated to be highly expressed during *E. pestis* infection [61], suggesting the activation of NF-κB that protects the infected cells from apoptosis and allows the maturation of the second generation of merozoites [7]. Consistently, when the p65 subunit was deleted, the host cells lost the inhibition of apoptosis during infection [62].

Cells infected with *C. parvum* activate the NF-κB signaling pathway and are accompanied by the secretion of interleukin-8 (IF-8). Inhibitors MG-132 and SN50 can inhibit the binding of NF-κB to DNA and increase the expression level of IF-8, but significantly promote the H69 cell apoptosis level after *C. parvum* infection. The results indicate that the cell apoptosis rate after *C. parvum* infection is influenced by NF-κB. The ability of *C. parvum* to induce cell apoptosis increases due to the inactivation of the NF-κB signaling pathway. HCT-8 cells apoptosis rates with *C. parvum* infection in vitro after adding pro-apoptosis drugs were tested. *C. parvum* significantly reduces the cell apoptosis rate induced by these drugs, indicating that *C. parvum* can inhibit the cell apoptosis induced by pro-apoptotic factors, which is beneficial for its own proliferation and development.

### 5.5. Phosphatidylinositol 3-Kinase-Akt Signaling Pathway

Phosphatidylinositol 3-kinase (PI3K) is a large family of lipid kinases, which can be divided into classes I, II, and III according to their structure [63]. Of them, class I is a heterodimer composed of the catalytic subunit (P110) and regulatory subunit (P85). Under normal conditions, the p85-p110 complex is ubiquitously localized in the cytoplasm. After P85 and P110 are activated, they combine to generate a heterodimer, which activates the Akt signaling pathway. Then, the activated Akt inhibits the expression of apoptotic genes and enhances the expression of anti-apoptotic genes via the direct phosphorylation of Forkhead family transcription factors or apoptosis cascade regulatory proteins, or via the upregulation of NF-κB [64]. In macrophages infected with *T. gondii*, PI3K was demonstrated to be activated by GiPCR signal, leading to PI3K/AKT and MEK1/2 phosphorylation and MEK1/2-mediated ERK1/2 activation. Moreover, *T. gondii* inhibited crisscrotidyline-induced apoptosis, while PI3K inhibitors LY294002 blocked the ability of *T. gondii* to inhibit apoptosis, indicating apoptosis inhibition via the PI3K-Akt signaling pathway [55].

After being bitten by infected mosquitoes, the traversal of host hepatocytes by *Plasmodium* sporozoites (transmission type) leads to the release of hepatocyte growth factor (HGF) and ultimately promotes the invasion of hepatocytes where the parasite will reside, differentiate, and reproduce [65]. PI3K participates in inhibiting liver cell apoptosis in the early stages of *Plasmodium* parasite infection. HGF/Met signaling inhibits cell apoptosis through PI3K signaling [66]. This inhibitory effect enhances the process of *Plasmodium* infecting cells, and the use of PI3K inhibitors can lead to cell apoptosis after infection. It has been reported that the overexpression of glutamylcysteine can inhibit TNF-α-induced cell apoptosis and regulate the activation of NF-κB during *Plasmodium* parasite inhibiting cell apoptosis. It is interesting that the inhibitory effect of infected liver cell apoptosis increases with infection time and shows resistance to cell apoptosis both in vivo and in vitro [67]. The inhibition of cell apoptosis by *Plasmodium* not only occurs during the replication stage of hepatocytes but also precise control over host cell apoptosis in the final stage of hepatocyte development. It was found that before erythrocyte infection, an increasing number of hepatocytes were filled with merozoites. These isolated hepatocytes exhibit various characteristics of cell apoptosis, including cytoplasmic cytochrome lysis, mitochondrial membrane potential loss, and nuclear coagulation, but lack phosphatidylserine in the outer layer of the plasma membrane. The results show that in the early stages of hepatocytes infection, the inhibition of cell apoptosis is dependent on the induction of non-apoptotic and non-necrotic cell death by merozoites, which promotes the formation of vesicles transporting merozoites, where their release leads to erythrocyte infection.

## 6. Conclusions

One of the main mechanisms by which Apicomplexan protozoa inhibit cell apoptosis is their ability to delay host cell damage. In the early stages of infection, Apicomplexan protozoa inhibit host cell apoptosis in order to ensure their own growth, development, and reproduction within the cell. The regulation mechanism of apoptosis is inhibited by the interaction between host cells and parasites, including main virulence factors associated with Apicomplexan protozoa that inhibit host cell apoptosis.

Through the study of molecular mechanisms, we have revealed some key factors and signaling pathways of apoptosis inhibition in host cells, which regulate apoptosis signal transduction, and molecular mechanisms that interfere with apoptosis. These findings provide important clues for further understanding the infection mechanism and pathogenesis of Apicomplexan protozoa and a theoretical basis for the development of new therapeutic strategies and preventive measures.

Although we have made some important progress, there are still many questions that need to be further explored. Future research can focus on the molecular mechanisms of apoptosis regulation, the detailed mechanisms of interactions between Apicomplexan protozoa and host cells, and the therapeutic strategies for parasite infections. In summary, studies on the inhibition of apoptosis by the interactions between parasite and host in the early stage of parasite infection provide important scientific evidence for us to understand the mechanism of parasitic infections, develop effective treatments, and prevent and control infectious diseases. In future studies, we will be able to better understand and respond to the challenges posed by Apicomplexan protozoa infections and contribute to human and animal health.

## Figures and Tables

**Figure 1 animals-13-03817-f001:**
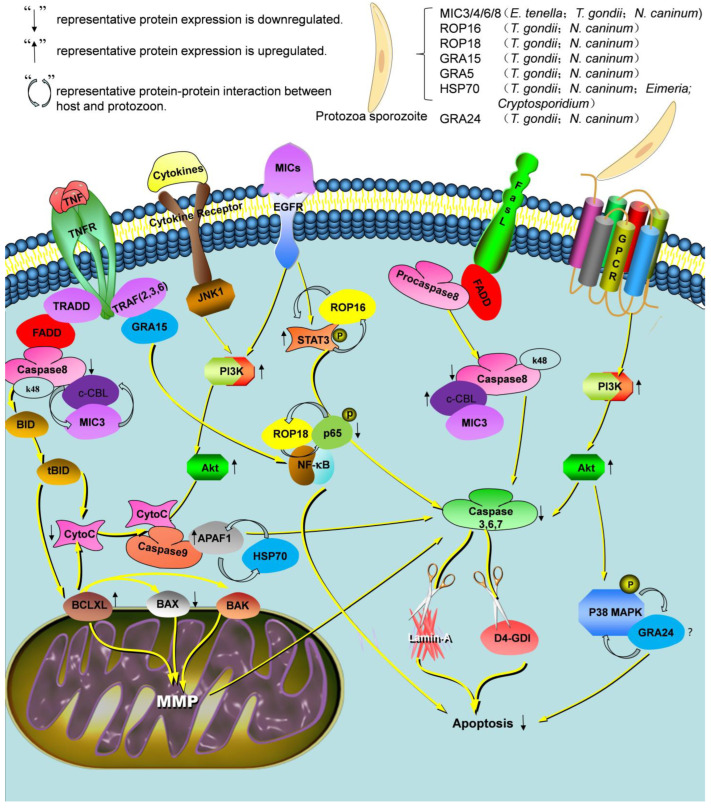
Signaling pathways involved in the inhibition of apoptosis by Apicomplexan protozoa. Schematic representation of the proposed apoptosis inhibition mechanisms of Apicomplexan protozoa in host cells. In this model, Apicomplexan protozoa inhibit EGFR, Jun N-terminal kinase (JNK), and p38 phosphorylation (proapoptotic mechanisms) signaling pathway, while activating phosphatidylinositol 3-kinase (PI3K)/Akt (antiapoptotic mechanisms). FADD, Fas-associating protein with a novel death domain; TRADD, TNF receptor-associated death domain; STAT3, signal transducer and activator of transcription 3; NF-κB, nuclear factor kappa-light-chain-enhancer of activated B cells; MMP, mitochondrial membrane potential; APAF-1, apoptotic protease activating factor-1; GPCR, G protein-coupled receptor.

**Table 1 animals-13-03817-t001:** The main proteins thus far described participate in the inhibition of apoptosis induced by Apicomplexan protozoa.

	Signaling Pathway	Apicomplexan Protozoa	References
MIC6	EGFR/PI3K/Akt	*T. gondii*	Muniz-Feliciano L et al. (2013) [30]
ROP16	JKT-STAT3	*T. gondii*	Yamamoto M et al. (2009) [33]
ROP18	NF-κB	*T. gondii*	Du J et al. (2014) [35]
GRA15	NF-κB	*T. gondii*	Sangaré LO et al. (2019) [43]
HSP70	Cytochrome-C	*T. gondii*	Mosser D et al. (2000) [48]
HSP70	APAF-1	*T. gondii*	Saleh A et al. (2000) [49]
MIC3	EGFR/PI3K/Akt	*Eimeria*	Muniz-Feliciano L et al. (2013) [30]
MIC3	CBL	*Eimeria*	Wang P et al. (2021) [6]
GRA5	Unknown	*T. gondii*	Ching, X.T et al. (2015) [40]
MIC8	EGFR/PI3K/Akt	*T. gondii*	Meissner M et al. (2002) [31]
MIC4	EGFR/PI3K/Akt	*Eimeria*	Xuesong Zhang et al. (2022) [25]

## Data Availability

No new data were created or analyzed in this study. Data sharing is not applicable to this article.

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
