# Peer review of "Inhibition of Cell Apoptosis by Apicomplexan Protozoa–Host Interaction in the Early Stage of Infection"

_animals, 2023, doi:10.3390/ani13243817_

Round 1
Reviewer 1 Report
Comments and Suggestions for Authors
This manuscript summarized and reviewed the inhibition of cell apoptosis by Apicomplexan protozoa. In my eyes, it deserves to press in the Journal (Honestly, although I understand the content of the manuscript, but I have not done related work in the field, Therefore, my review probably acts as a reference.
Suggestions
Line 93 the headline is “Apicomplexan protozoa inhibit cell apoptosoiis..”,. Leishmania and Trypanosoma are not apicomplexan protozoa. However, under the headline, Leishmania and Trypanosoma are not apicomplexan protozoa.
Minor questions
Line 95,spelling mistake Cryptosporid-iumspp,. Additionally, whether this genus belonging to apicomplexan protozoa should be considered based references published recent years.
Lines: 165-166, the abbreviation “RONs and ROPs” the I fell a little funny.
Line 169 JKT-STAT should use full spelling for its first mentioned.
Author Response
Suggestions
Line 93 the headline is “Apicomplexan protozoa inhibit cell apoptosis”, Leishmania and Trypanosoma are not apicomplexan protozoa. However, under the headline, Leishmania and Trypanosoma are not apicomplexan protozoa.
Answer: Thank you very much for your suggestion. Leishmania and Trypanosoma have been deleted, furthermore, Plasmodium has been added into the manuscript in red line “75-84”, “303-323”. Cryptosporidium has been added into the manuscript in red line “118-120”, ”252-263”.
Minor questions
Line 95,spelling mistake Cryptosporid-iumspp,. Additionally, whether this genus belonging to apicomplexan protozoa should be considered based references published recent years.
Answer: Thank you very much for your suggestion. “Cryptosporid-iumspp” has been revised into Cryptosporidium in red line 33. Apicomplexa protozoa including Toxoplasma gondii (T. gondii), Plasmodium, Cryptosporidium, Eimeria and et al, are obligatory intracellular parasites, often zoonotic pathogens, posing considerable threats to social development and public health. The references are blow.
- Sardinha-Silva A, Alves-Ferreira EVC, Grigg ME. Intestinal immune responses to commensal and pathogenic protozoa. Front Immunol. 2022 Sep 16;13: 963723. doi: 10.3389/fimmu.2022.963723.
- Boisard J, Florent I. Why the -omic future of Apicomplexa should include gregarines. Biol Cell. 2020 Jun;112(6):173-185. doi: 10.1111/boc.202000006.
- Sparvoli D, Lebrun M. Unraveling the Elusive Rhoptry Exocytic Mechanism of Apicomplexa. Trends Parasitol. 2021 Jul;37(7):622-637. doi: 10.1016/j.pt.2021.04.011.
Lines: 165-166, the abbreviation “RONs and ROPs” the I fell a little funny.
Answer: Thank you very much for your suggestion. The abbreviation “RONs and ROPs” has been revised into rhoptry neck proteins (RONs) and rhoptry bulb proteins (ROPs) in red line 129.
Line 169 JKT-STAT should use full spelling for its first mentioned.
Answer: Thank you very much for your suggestion. The full spelling of JKT-STAT is Janus Kinase-signal transducer and activator of transcription in red line 131-132.

Reviewer 2 Report
Comments and Suggestions for Authors
This manuscript highlights an interesting topic that enhances our understanding of host-pathogen interactions and merit further discussion. Overall, this manuscript is somewhat summarized, but some errors should be corrected before acceptance
Major comments:
1. The authors should clearly define the Apicomplexa protozoans that they are going to discuss. As far to my knowledge, at least, Leishmania or Trypanosoma should not be categorized as Apicomplexa and included in this discussion.
2. The authors almost only focus on T. gondii and Eimeria but did not mention about other important pathogenic Apicomplexa protozoan such as Plasmodium and Cryptosporidium. These parasites should not be ignored if they use the current title to conclude the contents.
3. The quality of Figure should be improved. It is hard to understand which or how the protozoan-derived factors interact with host molecules. Pathway network should be summarized more clearly.
4. Some description of the effect of inhibition of host cell apoptosis on parasite survival should be added according to previous reports.
Minor comments:
1. Line 118-120: the meaning of this sentence is unclear.
2. Give the full name of abbreviations when they are used at the first time or list them up.
3. Line 171-176: grammatical errors
4. Figure 1 is not only related to HSPs. Line 226-229 should not be included in HSPs part.
Comments on the Quality of English LanguageEnglish writing should be improved as some grammatical errors can lead to misunderstanding. The meaning of some sentences are vague.
Author Response
Major comments:
- The authors should clearly define the Apicomplexa protozoans that they are going to discuss. As far to my knowledge, at least, Leishmaniaor Trypanosoma should not be categorized as Apicomplexa and included in this discussion.
Answer: Thank you very much for your suggestion. Leishmania and Trypanosoma have been deleted in the manuscript.
- The authors almost only focus on gondiiand Eimeria but did not mention about other important pathogenic Apicomplexa protozoan such as Plasmodium and Cryptosporidium. These parasites should not be ignored if they use the current title to conclude the contents.
Answer: Thank you very much for your suggestion. Plasmodium has been added into the manuscript in red line “79-89”, “342-363”. Cryptosporidium has been added into the manuscript in red line “129-131”, “291-300”,“35-36”.
- The quality of Figure should be improved. It is hard to understand which or how the protozoan-derived factors interact with host molecules. Pathway network should be summarized more clearly.
Answer: Thank you very much for your suggestion. Fig. 1 has been redrawn in the manuscript. HSP70 in T. gondii has been added into Fig 1. It inhibits the cell apoptosis via cytochrome c release, anti-apoptotic protein (APAF-1) expression and caspase inhibition. E. acervulina MIC 3 protein can inhibits apoptosis of the chicken duodenal epithelial cell by interacting with the casitas B-lineage lymphoma (CBL) protein. The over-expression of E. acervulina MIC 3 protein increased the level of CBL expression, which mediated the ubiquitination of caspase 8 and promoted its degradation. The decrease of caspase 8 expression level finally led to the inhibition of cell apoptosis. MIC6 protein contains EGF motif, which can interact with EGFR and activate the EGFR-Akt signaling pathway, finally inhibit host cell apoptosis. GRA24 secreted into the cytoplasm and bound to p38 MAPK, activated the p38 translocation to the nucleus and activated p38 activation and PI3K/AKT signaling pathways, which inhibited apoptosis of infected cells. GRA15 and GRA 24 can regulate the cell apoptosis via the activation of NF-κB signaling pathways. GRA5 can inhibit apoptosis of the infected cells thereby protecting the parasites, but the mechanism is unclear.
- Some description of the effect of inhibition of host cell apoptosis on parasite survival should be added according to previous reports.
Answer: The effect of inhibition of host cell apoptosis on parasite survival has been added in red line “75-96”, “338-341”, “342-363”, “79-88”, “174-177”, “253-256”
Minor comments:
- Line 118-120: the meaning of this sentence is unclear.
Answer: During the early infection of Toxoplasma gondii, apoptosis suppression also results from decreasing the expression of pro-apoptotic proteins, such as Bax and Bak, and increasing the expression of anti-apoptotic proteins, thereby preventing the mitochondria-mediated cell apoptosis. The sentence has been revised in red line “97-100”
- Give the full name of abbreviations when they are used at the first time or list them up.
Answer: Toxoplasma gondii, T. gondii; epidermal growth factor receptor (EGFR); TNF receptor-associated death domain (TRADD); mitochondrial permeability transition pore (MPTP); microneme proteins (MICs); rhoptry proteins in the neck (RONs); rhoptry proteins in the bulb (ROPs); parasitophorous vacuole (PV); moving junction (MJ); nuclear localization structure (NLS); Dense granule proteins (GRA); Inhibitors of STAT transcription (IST). mitochondrial association factors (MAF); c-Myc regulation (MYR); parasitophorous vacuole membrane (PVM); heat shock proteins (HSPs); Jun N-terminal kinase (JNK); phosphatidylinositol 3-kinase (PI3K); Nuclear factor kappa (NF-κB); Immunoactive (IC) and Immunosuppressive (IS); silver nanoparticles (AgNPs); Interleukin-8 (IF-8); Hepatocyte growth factor (HGF); APAF-1, Apoptotic protease activating factor-1; FADD, Fas-associating protein with a novel death domain.
- Line 171-176: grammatical errors
Answer: “It was released in the cytoplasm and then translocated into the nucleus of host cells through its nuclear localization structure (NLS), thereby affecting the expression of related genes.” The sentence has been revised in red line “137-139”.
- Figure 1 is not only related to HSPs. Line 226-229 should not be included in HSPs part.
Answer: HSP70 has been added into Fig 1. Line 181-184 has been taken out as a separate paragraph.
HSP70 in T. gondii has been added into Fig 1. It inhibits the cell apoptosis via cytochrome c release, anti-apoptotic protein (APAF-1) expression and caspase inhibition. The study demonstrated that exertion of HSP70 and HSP90 anti-apoptotic effects are directly associated with pro-apoptotic factors APAF-1 and Apaf-1. HSP70 can bind to AIF (apoptosis-inducing factor) and apoptosis protease activating factor 1 (APAF-1), over-express anti-apoptotic Bcl-2 and then inhibit mitochondrial apoptosis
Reviewer 3 Report
Comments and Suggestions for Authors
Author Response
We have revised the manuscript according to your suggestion. Thank you very much for your help!

Reviewer 4 Report
Comments and Suggestions for Authors
The purpose of this review is to describe how Apicomplexan parasites inhibit apoptosis of their host cell.
The topic of this review is interesting but there are several major flaws.
1. There is a lot of redundancy within sections (multiple sentences that state the same point) and between sections.
2. Another major flaw is in the characterization of protozoa. In the title and abstract, the authors focus on the phylum Apicomplexa. However, in the Introduction and following sections, the authors describe work on Leishmania and Trypanosoma. Neither of these genera are in the phylum Apicomplexa but are rather in the phylum Euglenozoa.
3. There is not enough detail in any of the sections. For example, in section 3 (Apicomplexan protozoa inhibit cell apoptosis in the early stage of infection) the authors state that Leishmania can interfere with mitochondrial membrane potential but there are no details as to how. What proteins are involved? What is known in the field? In section 4 (Virulence factors associated with Apicomplexan protozoa inhibit host cell apoptosis), there is not enough detail to understand the role of MIC3, MIC6 and MIC8 in inhibiting host cell apoptosis. Which host components do these parasite proteins target? Since these proteins are involved in promoting invasion and inhibiting apoptosis, how are these two roles regulated? In the section on dense granule proteins, it is confusing as to when the authors are describing dense granule organelles versus dense granule proteins.
Comments on the Quality of English Language
There were a number of sentences that were confusing and would need to be rewritten. For example, on lines 83-84 ("Activated aptamers further recruit and activate protein kinases, such as members of the caspase protein family") makes it seem that caspase family members are kinases instead of proteases that may regulate kinases.
Author Response
The topic of this review is interesting but there are several major flaws.
- There is a lot of redundancy within sections (multiple sentences that state the same point) and between sections.
Answer: Thank you very for your suggestion. The manuscript has been rewritten. The redundant sentences have been deleted as much as possible.
- Another major flaw is in the characterization of protozoa. In the title and abstract, the authors focus on the phylum Apicomplexa. However, in the Introduction and following sections, the authors describe work on Leishmania and Trypanosoma. Neither of these genera are in the phylum Apicomplexa but are rather in the phylum Euglenozoa.
Answer: Thank you very much for your suggestion. Leishmania and Trypanosoma have been deleted in the manuscript. Plasmodium has been added into the manuscript in red line “79-88”, “342-363”. Cryptosporidium has been added into the manuscript in red line “129-131”, ”291-300”.
- There is not enough detail in any of the sections. For example, in section 3 (Apicomplexan protozoa inhibit cell apoptosis in the early stage of infection) the authors state that Leishmania can interfere with mitochondrial membrane potential but there are no details as to how. What proteins are involved? What is known in the field? In section 4 (Virulence factors associated with Apicomplexan protozoa inhibit host cell apoptosis), there is not enough detail to understand the role of MIC3, MIC6 and MIC8 in inhibiting host cell apoptosis. Which host components do these parasite proteins target? Since these proteins are involved in promoting invasion and inhibiting apoptosis, how are these two roles regulated? In the section on dense granule proteins, it is confusing as to when the authors are describing dense granule organelles versus dense granule proteins.
Answer:
- Thank you very much for your suggestion.Descriptions of Leishmania have been deleted in section 3. The roles of MIC3, MIC6 in inhibiting host cell apoptosis have been described in red line “118-125”.
“E. tenella MIC4 protein contains EGF motif, which can interact with EGFR and activate the EGFR-Akt signaling pathway, finally inhibit host cell apoptosis[31]. Similarly, E. acervulina MIC 3 protein can inhibits apoptosis of the chicken duodenal epithelial cell by interacting with the casitas B-lineage lymphoma (CBL) protein. The over-expression of E. acervulina MIC 3 protein increased the level of CBL expression, which mediated the ubiquitination of caspase 8 and promoted its degradation. The decrease of caspase 8 expression level finally led to the inhibition of cell apoptosis”
- MICs, RONs, ROPs, GRAs and HSPs proteins mediate the invasion of apicomplexan protozoa, and then increase the number of cells infected with parasites. Apicomplexa protozoa including Toxoplasma gondii (T. gondii), Plasmodium, Cryptosporidium, Eimeriaand et al, can inhibit cell apoptosis in order to its intracellular long-term parasitism. Therefore, the number of Apicomplexan protozoa in host cells is closely related with cell apoptosis. The references are as follows:
Guillermo, L.V., et al., Targeting caspases in intracellular protozoan infections. Immunopharmacol Immunotoxicol, 2009. 31(2): p. 159-73.
Heussler VT, Küenzi P, Rottenberg S. Inhibition of apoptosis by intracellular protozoan parasites. Int J Parasitol. 2001. 31(11):1166-76.
Gervais, O., T. Renault, and I. Arzul, Molecular and cellular characterization of apoptosis in flat oyster a key mechanisms at the heart of host-parasite interactions. Sci Rep, 2018. 8(1): p. 12494.
Bosurgi L, Rothlin CV. Management of cell death in parasitic infections. Semin Immunopathol. 2021. 43(4):481-492.
- The dense granule proteins on section 4 have been rewritten in red line “153-164”
“GRAs related to apoptosis were observed in Toxoplasma-infected cells, such as mitogen-activated protein (MAP) kinase (GRA24), NF-κB (GRA15), and P53 (GRA16. GRAs can inhibit apoptosis to achieve immune evasion and long-term parasitism. It was shown that IST (A dense granule protein) secreted by T. gondii crossed the parasitophorous vacuole membrane (PVM) and accumulated in the nucleus to block the type I interferon response and promoted the infection. GRA5 can inhibit apoptosis of the infected cells thereby protecting the parasites. Similarly, GRA24 secreted into the cytoplasm and bound to p38 MAPK, activated the p38 translocation to the nucleus and activated p38 activation and PI3K/AKT signaling pathways, which inhibited apoptosis of infected cells.”
Comments on the Quality of English Language
There were a number of sentences that were confusing and would need to be rewritten. For example, on lines 83-84 ("Activated aptamers further recruit and activate protein kinases, such as members of the caspase protein family") makes it seem that caspase family members are kinases instead of proteases that may regulate kinases.
Answer: The lines 83-84 ("Activated aptamers further recruit and activate protein kinases, such as members of the caspase protein family") have been revised in the manuscript. The sentence has been revised in red line 66-72 “The process includes the following steps: first, ligands (such as tumor necrosis factor-α, Fas ligand) bind to the corresponding receptors, resulting in the aggregation of receptors and the formation of aggregation areas, a key step in the apoptotic signaling downstream signaling pathways are activated by binding to appropriate aptamers, which typically include intracellular death proteins that interact with the extracellular domain of the receptor, leading to recruitment and activation of protein kinases, such as members of the caspase protein family.” The entire manuscript has been rewritten and the added paragraphs and sentences are marked in red.
Round 2
Reviewer 2 Report
Comments and Suggestions for Authors
This revised version has overall improved compared to the previous manuscript. However, it still need further refinement.
Firstly, I believe the content written in the text is not effectively summarized in the figure. While the figure includes pathways related to apoptosis, it does not clearly depict the specific molecules derived from which parasites act on which factors in the pathways. As a result, the figure doesn't seem to have much value as a review figure. It would be desirable to provide a more detailed summary focusing on the movement of each molecule.
Secondly, the information compiled in the table is less extensive than what is presented in the figure. It might be good to double-check if there is any omissions and ensure that the table encompasses all the relevant details discussed in the figure.
Comments on the Quality of English Languagethe English writing has improved and is readable.
Author Response
Reviewer 1: This revised version has overall improved compared to the previous manuscript. However, it still need further refinement.
Firstly, I believe the content written in the text is not effectively summarized in the figure. While the figure includes pathways related to apoptosis, it does not clearly depict the specific molecules derived from which parasites act on which factors in the pathways. As a result, the figure doesn't seem to have much value as a review figure. It would be desirable to provide a more detailed summary focusing on the movement of each molecule.
Answer: Thank you very much for your help. The figure has been re-revised according to your suggestion. Protozoa inhibiting host cell apoptosis through parasites proteins interaction with host proteins has been demonstrated as much as possible. The relevant signaling pathways of apoptosis inhibition have been marked in the figure, and the up-regulation or down-regulation of related host proteins expression has been shown in the figure.
Secondly, the information compiled in the table is less extensive than what is presented in the figure. It might be good to double-check if there is any omissions and ensure that the table encompasses all the relevant details discussed in the figure.
Answer: Thank you very much for your help. There are a total of 10 proteins introduced in the picture and table. At present, it has been reported that inhibition of cell apoptosis related proteins is mainly found in Eimeria and Toxoplasma gondii, but homologous proteins are also present in Neosporidium and Cryptosporidium. The homologous proteins of different protozoa were shown in figure and the inhibition of cell apoptosis related pathways in Eimeria and Toxoplasma gondii were shown in Table.
Reviewer 4 Report
Comments and Suggestions for Authors
In the Introduction, what do the authors mean by “considerable threats to social
development”? (Line 27)
Line 59: remove “such as activated cascade family members” as it is redundant in the
sentence. It would be helpful to include an example or two of other family members. For
example, the sentence can be the following: “thirdly, the activated caspase-9 further activates
other caspase family members (e.g., “give a few examples”), that initiate a cascade in the cells
that cleaves specific intracellular proteins (e.g., “give a few examples”), ultimately leading to
the death of the cell.
Line 61-61: The word “unlikely” means improbable or not likely to happen. It should be deleted
Lines 63-69: The added sentence is confusing and needs to be rewritten. Is this meant to be
two sentences? In particular, “the apoptotic signaling downstream signaling pathways” (line 66)
is particularly confusing.
Lines 71-72: Rewrite, “Eventually, cells fragment into apoptotic bodies that are…”
Lines 76-85: In the first sentence, the authors mention that Plasmodium inhibits host cell apoptosis and mention caspase activity, the PI3K signaling pathway, and the regulation of apoptosis- related protein expression. No details are provided on how Plasmodium inhibits the host pathways and activities.
The authors then state that traversal of Plasmodium sporozoites cause the release of
HGF. Is this related in any way to the 3 host targets mentioned in the first sentence. Then, the
authors state that “infected hepatocytes express HGR receptor c-met, which ensures the initial
survival of the cells” but there is not mention of how this helps the cells to survive.
In lines 81-85, the authors should make it clear that apoptosis is induced in mammalian
cells in vitro by peroxide treatment or serum deprivation but in infected cells the parasite can
inhibit this induction.
Lines 96-97: It would be helpful to list an example or two of anti-apoptotic proteins whose
expression is increased upon Toxoplasma infection.
Line 111: Traditionally, micronemes are referred to in plural because there are multiple
micronemes in Apicomplexan parasites.
Line 134: I recommend modifying this sentence. For the part “thereby affecting the expression
of related genes” change “related genes” to “host genes”. It is unclear what is meant by related genes as the structure of the sentence suggests genes related to ROP16.
Lines 140-150: “GRAs related to apoptosis were observed in Toxoplasma-infected, cells such as mitogen-activated protein (MAP) kinase (GRA24)….” The way this sentence is written seems to suggest that GRA24 is a MAP kinase but it is not. Instead, GRA24 interacts with host MAP kinase, which is made clearer later in the text. So, instead rewrite the sentence to just list the parasite proteins that are related to host apoptosis pathways.
Lines 154-155: Is anything known about how GRA5 inhibits apoptosis? If not, then it would be helpful for the reader to state that the mechanism is as yet unknown.
Line 169: It would be helpful to list a few examples of anti-apoptotic proteins whose expression is affected by Toxoplasma HSP70 or HSP90. Plus, how is the expression changed? Upregulated or downregulated?
Lines 262-271: This paragraph could use some editing of the English.
Lines 311-314
The transmission type of Plasmodium is sporozoites and not spores. The sporozoites invade hepatocytes but not specifically daughter hepatocytes (delete the word daughter).
The authors write that the sporozoites invade hepatocytes and develop within them leading to the release of HGF that promotes the invasion of hepatocytes where the parasite reside, differentiate and reproduce. Do the authors mean to describe the traversal of host hepatocytes by the parasite that leads to the release of HGF and promotes invasion with further development of the parasites (differentiation and reproduction)?
Comments on the Quality of English Language
It would be useful to have the paper edited for English.
Author Response
Reviewer 2: In the Introduction, what do the authors mean by “considerable threats to social development”? (Line 27)
Answer: Thank you very much for your help. The sentence has been revised into “considerable threats to economic development and public health”.
Line 59: remove “such as activated cascade family members” as it is redundant in the sentence. It would be helpful to include an example or two of other family members. For example, the sentence can be the following: “thirdly, the activated caspase-9 further activates other caspase family members (e.g., “give a few examples”), that initiate a cascade in the cells that cleaves specific intracellular proteins (e.g., “give a few examples”), ultimately leading to the death of the cell.
Answer: Thank you very much for your help. The sentence has been revised in red line “thirdly, the activated caspase-9 further activates other caspases family members (caspase-7, caspase-6 and caspase-3), initiates a cascade in the cell that cleaves specific intracellular proteins (PARP (PolyADP-Ribose Polymerase), GDID4, DFF-45 (DNA fragmentation factor-45), lamin A, keratin 18), ultimately leads to the death of the cell”.
Line 61-61: The word “unlikely” means improbable or not likely to happen. It should be deleted
Answer: Thank you very much for your help. The word “unlikely” has been deleted.
Lines 63-69: The added sentence is confusing and needs to be rewritten. Is this meant to be two sentences? In particular, “the apoptotic signaling downstream signaling pathways” (line 66) is particularly confusing.
Answer: Thank you very much for your help. The sentence has been revised in red line “The exogenous pathway is a process of cell apoptosis mediated by transmembrane receptors such as such as tumor necrosis factor-α (TNF-α), Fas ligand. When there are corresponding ligand molecules outside the cell, the receptor proteins on the cell membrane are recruited together and bind to the ligand, forming a death inducing signal complex, catalyzing and activating procaspase-8, finally leading to the executive phase of cell apoptosis is activated”.
Lines 71-72: Rewrite, “Eventually, cells fragment into apoptotic bodies that are…”
Answer: Thank you very much for your help. The sentence has been revised in red line “Eventually, cells fragment into apoptotic bodies that are taken up and phagocytosed by surrounding macrophages to complete the process of cell clearance”.
Lines 76-85: In the first sentence, the authors mention that Plasmodium inhibits host cell apoptosis and mention caspase activity, the PI3K signaling pathway, and the regulation of apoptosis- related protein expression. No details are provided on how Plasmodium inhibits the host pathways and activities.
Answer: Thank you very much for your help. PI3K participates in inhibiting liver cell apoptosis in the early stages of Plasmodium infection. This inhibitory effect enhances the process of Plasmodium infecting cells, and the use of PI3K inhibitors can lead to cell apoptosis after infection. Inhibition of PI3-kinase pathway decreases Plasmodium infection in vivo and causes a higher level of apoptosis in Plasmodium-infected host cells. PI3K is activated and converted from PIP3 to PI(3)P through phosphorylation. Hsp70-1 is considered as a PI(3)P-binding protein. An Hsp70 inhibitor and knockdown of Hsp70-1 phenocopy PI(3)P-deficient parasites under heat shock. PI(3)P protect Plasmodium falciparum from heat-induced cell death, but the specific mechanism is currently unclear. The reference is below:
Lu KY, Pasaje CFA, Srivastava T, Loiselle DR, Niles JC, Derbyshire E. Phosphatidylinositol 3-phosphate and Hsp70 protect Plasmodium falciparum from heat-induced cell death. Elife. 2020, 25;9:e56773. doi: 10.7554/eLife.56773.
Leirião P, Albuquerque SS, Corso S, van Gemert GJ, Sauerwein RW, Rodriguez A, Giordano S, Mota MM. HGF/MET signalling protects Plasmodium-infected host cells from apoptosis. Cell Microbiol. 2005, 7(4):603-9.
The authors then state that traversal of Plasmodium sporozoites cause the release of HGF. Is this related in any way to the 3 host targets mentioned in the first sentence. Then, the authors state that “infected hepatocytes express HGR receptor c-met, which ensures the initial survival of the cells” but there is not mention of how this helps the cells to survive.
Answer: Thank you very much for your help. The sentence has been revised in red line 78-80 “Since HGF/MET signalling is capable of protecting cells from apoptosis by using both PI3-kinase/Akt and MAPK pathways, inhibition of HGF/MET signalling induces a specific increase in apoptosis of infected cells leading to a great reduction on infection. HGF/MET signalling protects infected host cells via activation of PI3-kinase pathway.”
In lines 81-85, the authors should make it clear that apoptosis is induced in mammalian cells in vitro by peroxide treatment or serum deprivation but in infected cells the parasite can inhibit this induction.
Answer: Thank you very much for your help. The sentence has been revised in red line “The apoptosis is induced in mammalian cells in vitro by peroxide treatment or serum, but Plasmodium can inhibit cell apoptosis in vitro by peroxide treatment or serum deprivation and in vivo by TNF-α.”
Lines 96-97: It would be helpful to list an example or two of anti-apoptotic proteins whose expression is increased upon Toxoplasma infection.
Answer: Thank you very much for your help. The sentence has been revised in red line “Toxoplasma GRA24 can induced higher activation of the subunit p38-alpha. GRA15 interacts with TNF receptor-associated factors (TRAFs), which are adaptor proteins functioning upstream of the NF-κB transcription factor. Deletion of TRAF-binding sites greatly reduces its ability to activate the NF-κB pathway, and TRAF2 knockout cells have impaired GRA15-mediated NF-κB activation.”
Line 111: Traditionally, micronemes are referred to in plural because there are multiple micronemes in Apicomplexan parasites.
Answer: Thank you very much for your help. The sentence has been revised in red line “Micronemes are a special organelle located at the anterior end of apicomplexan.”
Line 134: I recommend modifying this sentence. For the part “thereby affecting the expression of related genes” change “related genes” to “host genes”. It is unclear what is meant by related genes as the structure of the sentence suggests genes related to ROP16.
Answer: Thank you very much for your help. The sentence has been revised in red line “It was released in the cytoplasm and then translocated into the nucleus of host cells through its nuclear localization structure (NLS), thereby affecting the expression of host genes.”
Lines 140-150: “GRAs related to apoptosis were observed in Toxoplasma-infected, cells such as mitogen-activated protein (MAP) kinase (GRA24)….” The way this sentence is written seems to suggest that GRA24 is a MAP kinase but it is not. Instead, GRA24 interacts with host MAP kinase, which is made clearer later in the text. So, instead rewrite the sentence to just list the parasite proteins that are related to host apoptosis pathways.
Answer: Thank you very much for your help. The sentence has been revised in red line “GRAs related to apoptosis were observed in Toxoplasma-infected cells, such as GRA24 and GRA15. Toxoplasma GRA24 can induced higher activation of the subunit p38-alpha MAPK. GRA15 interacts with TNF receptor-associated factors (TRAFs), which are adaptor proteins functioning upstream of the NF-κB transcription factor. Deletion of TRAF-binding sites in GRA15 greatly reduces its ability to activate the NF-κB pathway, and TRAF2 knockout cells have impaired GRA15-mediated NF-κB activation.”
Lines 154-155: Is anything known about how GRA5 inhibits apoptosis? If not, then it would be helpful for the reader to state that the mechanism is as yet unknown.
Answer: Thank you very much for your help. The sentence has been revised in red line “T. gondii GRA5 deletion mutant protects hosts against Toxoplasma gondii infection, but the mechanism is as yet unknown.”
Line 169: It would be helpful to list a few examples of anti-apoptotic proteins whose expression is affected by Toxoplasma HSP70 or HSP90. Plus, how is the expression changed? Upregulated or downregulated?
Answer: Thank you very much for your help. T. gondii infection induced overexpression of Bcl-2 and HSP70, inhibited the mitochondrial permeability transition pore, blocked release of both cytochrome c and AIF from mitochondria, and thus inhibited apoptosis. T. gondii infection-induced HSP70 interferes with apoptosis through the AIF- and cytochrome c-mediated programmed cell death pathway. HSP70 involves in the T. gondii infection-induced anti-apoptotic effect of Bcl-2. After release of cytochrome c mitochondria, it interacts with Apaf-1 in the presence of ATP. This interaction provokes a conformational change in Apaf-1 that recruits and activates procaspase-9. In T. gondii-infected cells, HSP70 bound to AIF and Apaf-1, and so overexpression of HSP70 inhibited AIF translocation and cytochrome c redistribution.
Lines 262-271: This paragraph could use some editing of the English.
Answer: Thank you very much for your help. The sentence has been revised in red line 259-269.
Lines 311-314
The transmission type of Plasmodium is sporozoites and not spores. The sporozoites invade hepatocytes but not specifically daughter hepatocytes (delete the word daughter).
Answer: Thank you very much for your help. The sentence has been revised in red line 310-312.
The authors write that the sporozoites invade hepatocytes and develop within them leading to the release of HGF that promotes the invasion of hepatocytes where the parasite reside, differentiate and reproduce. Do the authors mean to describe the traversal of host hepatocytes by the parasite that leads to the release of HGF and promotes invasion with further development of the parasites (differentiation and reproduction)?
Answer: Yes. The sporozoites cross the sinusoidal wall and migrate through several hepatocytes before they infect a final hepatocyte, with the formation of a parasitophorous vacuole, in which the intrahepatic form of the parasite grows and multiplies. The sentence has been revised in red line “the traversal of host hepatocytes by the parasite that leads to the release of HGF and promotes invasion with further development of the parasites (differentiation and reproduction).”